# Comparison of Efficacy and Inflammatory Response to Thermoconjunctivoplasty Performed with Cautery or Pulsed 1460 nm Laser

**DOI:** 10.3390/ijms24065740

**Published:** 2023-03-17

**Authors:** Rodrigo Guimaraes de Souza, David Huang, Scott Prahl, Lauren Nakhleh, Stephen C. Pflugfelder

**Affiliations:** 1Department of Ophthalmology, Baylor College of Medicine, Houston, TX 77030, USA; 2Department of Ophthalmology, Oregon Health Sciences Center, Portland, OR 97239, USA; 3Department of Electrical Engineering and Renewable Energy, Oregon Institute of Technology, Wilsonville, OR 97070, USA

**Keywords:** conjunctivochalasis, laser, near-infrared, thermocautery, thermoconjunctivoplasty

## Abstract

Conjunctivochalasis is a degenerative condition of the conjunctiva that disrupts tear distribution and causes irritation. Thermoreduction of the redundant conjunctiva is required if symptoms are not relieved with medical therapy. Near-infrared laser treatment is a more controlled method to shrink the conjunctiva than thermocautery. This study compared tissue shrinkage, histology, and postoperative inflammation in thermoconjunctivoplasty performed on the mouse conjunctiva using either thermocautery or pulsed 1460 nm near-infrared laser irradiation. Three sets of experiments were performed on female C57BL/6J mice (*n* = 72, 26 per treatment group and 20 control) to assess conjunctival shrinkage, wound histology, and inflammation 3 and 10 days after treatment. Both treatments effectively shrunk the conjunctiva, but thermocautery caused greater epithelial damage. Thermocautery caused greater infiltration of neutrophils on day 3 and neutrophils and CD11b^+^ myeloid cells on day 10. The thermocautery group had significantly higher conjunctival expression of *IL-1β* on day 3. Expression of chemokine CCL2 was higher in the conjunctiva on day 3 and tear concentrations were higher on day 7 in the laser group. These results suggest that pulsed laser treatment causes less tissue damage and postoperative inflammation than thermocautery while effectively addressing conjunctivochalasis.

## 1. Introduction

Conjunctivochalasis is a degenerative condition of the conjunctiva. It is defined as loose, non-edematous inferior bulbar conjunctiva [1]. The condition typically occurs bilaterally, and can involve the medial, lateral, or central portions of the inferior bulbar conjunctiva. As the conjunctiva becomes loose and folded with age, it can mechanically obstruct the inferior tear meniscus, alter tear distribution, and cause chronic irritation, tearing, and visual impairment [2]. The redundant conjunctiva disrupts functions of the tear meniscus, which serves as the reservoir for tears, delivers tears to the ocular surface during blinking, and carries tears to the lacrimal puncta for disposal. Age is the most important risk factor, with a steady increase in prevalence of conjunctivochalasis after the third decade of life [3].

Conjunctivochalasis severity is graded by a scheme proposed by Meller and Tseng [1]: grade 0 (no persistent fold); grade 1 (a small, single fold); grade 2 (two or more folds and below the tear meniscus); grade 3 (multiple folds and higher than the tear meniscus) [1,3]. Patients diagnosed with grade 2 disease or higher generally have clinically significant conjunctivochalasis and may require pharmacotherapy or surgery to reduce symptoms. Grade 3 may be further divided into cases that can be treated by cautery or the most severe cases that require excisional surgery [4]. It has been found that the severity of the grade is significantly lower in the central conjunctiva than in the nasal or temporal zones and that the overall grade is significantly higher in females [3].

An objective measure to quantify severity is anterior segment optical coherence tomography (AS-OCT) [5,6,7]. AS-OCT is a useful and reproducible tool to measure the cross-sectional area of conjunctiva prolapsing into the tear meniscus and to monitor the effectiveness of thermoreduction [7].

While conjunctivochalasis is a common cause of tear dysfunction and eye irritation, it typically does not respond to standard dry eye treatments such as artificial tears, punctal plugs, and anti-inflammatory agents [4]. The goal of surgical treatment of conjunctivochalasis is to remove the redundant tissue. Several techniques have been described, including elliptical excision, excision and amniotic membrane transplant, thermocautery, electrocautery, and a paste–pinch–cut technique [8,9,10,11,12]. In thermocautery and electrocautery, the epithelium and stroma of the conjunctiva are burnt and reduced. Cautery procedures can be painful and induce significant inflammation which can take weeks to a month to fully heal. There is a risk of damaging adjacent tissues during the procedure, as well as developing a pyogenic granuloma during healing that may require further treatment. The paste–pinch–cut approach removes the redundant tissue, then reapproximates and seals the wound edges with fibrin sealant. Although this approach is less painful, it must be performed in an operating room and it also requires several weeks for recovery [11,12].

Lasers have been reported as being capable of delivering precise amounts of energy to shrink redundant areas and minimize trauma to the epithelium and underlying stroma. Argon lasers can shrink the conjunctiva, but the light delivered at 532 nm wavelength is strongly absorbed by hemoglobin, causing vessel occlusions and ruptures [13]. Yang and Choi reported significant reduction of the conjunctivochalasis grade six months after argon laser treatment of the inferior bulbar conjunctiva, but patients were observed to have visible conjunctival hemorrhages 1 week after the procedure. In a previously reported study by Yang et al., a near-infrared 1460 nm wavelength handheld laser was used ex vivo on porcine eyes and achieved 45% conjunctiva shrinkage [14]. Light at 1460 nm is strongly absorbed by water and hemoglobin absorption is minimal. The purpose of this study is to compare tissue shrinkage, histology of the treatment area, and postoperative inflammation in thermoconjunctivoplasty performed on mouse conjunctiva using either thermocautery or 1460 nm wavelength near-infrared laser light.

## 2. Results

The effects on conjunctival shrinkage, wound histology, and inflammation after thermoconjunctivoplasty performed on the mouse conjunctiva by NIR laser or thermocautery were compared.

A.Conjunctival shrinkage

Both treatments effectively shrunk the conjunctiva (*p* = 0.0001 vs. pretreatment, Figure 1) and there was no difference between treatment groups.

B. Wound Histology

Histology was performed following treatment with NIR laser or thermocautery to evaluate the acute effects of treatment on the epithelium and underlying stroma. Three days following treatment, the epithelium in the conjunctiva and adjacent peripheral cornea was thinned but intact in the treatment zone (arrow) in all eyes treated with the laser, while it was absent in all eyes treated with cautery (arrow) in sections stained with H&E (Figure 2, top row). No fibrosis was noted in either treatment group in Masson Trichrome stained sections (Figure 2, bottom row). These findings indicate that the laser causes less damage to the conjunctival and adjacent corneal epithelium than cautery and that no fibrosis is noted at this timepoint.

C. Inflammatory Cell Infiltration

Whole mount conjunctivas were immunostained on D3 and D10 following treatment for neutrophils (Ly6G), CD11b^+^ myeloid (macrophage/monocyte) cells, and monocyte lineage cells (Ly6C) (Figure 3). Mean fluorescent intensity and cell number were measured in each group (Figure 4). The cautery treated group had a significantly greater number and sum intensity of Ly6G^+^ neutrophils on D3 and D10 and of CD11b^+^ cells on D10. These findings suggest that the cautery stimulates greater infiltration of the conjunctiva by innate immune cells.

D. Tear Cytokines

A panel of 12 cytokine/chemokines was measured in control eyes at D0 and treatment eyes at D0, D1, D3, and D7 by multiplex immunobead assay (Figure 5). There was no difference between groups at any time point in tear concentrations of IFN-γ, IL-1β, TNFα, IL-10, IL12p40 and p70, IL-17, IP-10 (CXCL10), and MIG (CXCL9). The IL-13 concentration significantly decreased from D0 to D1 in the laser group and significantly increased from D1 to D3 in the cautery group. CCL-2 was significantly higher in laser D7 compared to laser D0, laser D1, and cautery D7. VEGF was significantly higher in the D1 and D7 cautery and laser groups compared to their pretreatment (D0) values.

E. Conjunctival Gene Expression

Expression of inflammatory and fibrotic genes was measured by RT-PCR in the conjunctiva on D3 and D10 and compared to the untreated control (Figure 6). Expression of most genes was similar between treatment groups, except for *IL-1β*, which was significantly higher in the cautery vs. laser group on D3, and *CCL2*, which was significantly higher in the laser group vs. cautery on D3.

## 3. Discussion

This study compared efficacy, histology, and post-operative inflammatory markers of thermoconjunctivoplasty performed on mouse conjunctiva by two different methods, thermocautery or a pulsed near-infrared laser. Thermocautery and high frequency radiowave electrocautery are the most commonly utilized methods to surgically treat conjunctivochalasis in a clinic setting [8,9,10]. Energy delivered to tissue from these devices can exceed 1000 °F, which burns tissue, can cause an unpleasant odor, and can ignite a fire [15]. Thermocautery also carries the risk of inadvertently burning the cornea or lower lid if the patient blinks or the eye moves. NIR laser energy is delivered in pulses which can be regulated, and this wavelength is primarily absorbed by water [14]. Lower temperatures are generated by the laser than cautery, and damage only occurs in the area where the beam is focused [15]. These differences make the laser a safer option to shrink tissue.

Laser energy was delivered in pulses from this prototype NIR laser. Four laser pulses produced similar amounts of conjunctival shrinkage as thermocautery touching the tissue. However, thermocautery caused greater epithelial damage. No epithelium was seen on post treatment day three in the thermocautery group, while it was present in all laser treated eyes at the same time point. This indicates that equivalent laser-induced conjunctival shrinkage can occur without epithelial trauma. Histology was performed to evaluate acute effects of the treatments on the epithelium and stroma. There was no histological evidence of fibrosis 3 days following NIR laser or thermocautery, but this may not be sufficient time for a fibrotic response to be detected in tissue section. It is possible fibrosis could be detected at a later post treatment time point.

Treated eyes were evaluated for evidence of postoperative inflammation by measuring the number of inflammatory cells infiltrating the conjunctiva, concentrations of inflammatory mediators in tears, and expression of inflammatory genes in the conjunctiva. The thermocautery group had greater conjunctival infiltration of neutrophils on day three and both neutrophils and CD11b^+^ myeloid cells on day 10. Consistent with this finding, the thermocautery group also had significantly higher conjunctival expression of the proinflammatory cytokine IL-1β on day 3. CD11b is a protein integrin subunit found on innate inflammatory cells, including monocytes and macrophages [16]. IL-1β is produced by both epithelial and immune cells and functions as an important mediator of inflammation. It is induced in response to inflammatory stimuli and amplifies the inflammatory cascade through stimulating the production of other cytokines, chemokines, proteases, and vascular adhesion molecules [17]. Elevated *IL-1β* expression suggests the thermocautery group had greater post-treatment inflammation when compared to the laser therapy.

Expression of the chemokine *CCL2* was higher in the conjunctiva of the laser group on day 3, while tear concentrations were higher on day 7. CCL2 is produced by both the epithelium and inflammatory cells and serves as a monocyte chemoattractant to sites of infection and wound healing [18]. The higher concentration of this cytokine in the laser group may be due to preservation of the epithelium in the laser treated eyes.

Overall, the findings of these studies demonstrate that NIR causes less postoperative inflammation and tissue damage compared to thermocautery. Conjunctivochalasis is a prevalent disease that is typically underdiagnosed either because it is considered a normal finding, particularly in older patients, or it is not considered to cause of chronic ocular irritation. Indeed, conventional dry eye therapies can temporarily reduce irritation, tearing, and blurred vision symptoms caused by conjunctivochalasis, but they do not provide a permanent solution. Patients with Grade 3 conjunctivochalasis who do not respond to topical therapies require thermal or surgical reduction of the redundant conjunctiva to treat their symptoms. Successfully performed thermoreduction can cause complete resolution of symptoms and dramatically improve function and quality of life. While thermocautery is effective in shrinking the redundant conjunctiva, it is not always well tolerated by patients because they can experience discomfort and inflammation that can persist for several weeks. Occasionally pyogenic granuloma and fibrosis develop that require additional treatments. Some clinicians are reluctant to use thermocautery for this condition because of the risk of burning the surrounding tissues, such as the lid margins or cornea. It is more difficult to titrate thermocautery than a pulsed laser.

Compared to 514.5 nm argon green laser light that is absorbed by hemoglobin, 1460 nm NIR light is primarily absorbed by water and heating is confined to the upper 400 um of tissue [19]. The 200 ms pulse duration was chosen to minimize diffusion of heat from the treatment zone. The findings of our study suggest that NIR laser may be a safer alternative to thermocautery and provide the rationale for investigating the effectiveness and safety of this technology in future clinical trials.

A weakness of this study is that only the short-term effects of the two treatment modalities on the conjunctiva were evaluated. This model was well suited for comparing the relative proinflammatory effects of these treatments. Longer term evaluation of treated eyes is needed to assess durability of the treatment effect and potential complications, such as exuberant fibrosis. Use of an animal with eyes closer in size to the human may be more appropriate for these studies.

## 4. Materials and Methods

This protocol was approved by the Baylor College of Medicine Center for Comparative Medicine and conformed to the standards in the ARVO Statement for Use of Animals in Ophthalmic and Vision Research. Female C57BL/6J mice were purchased from Jackson Laboratories (Bar Harbor, Maine) and housed in the vivarium until use. Three sets of experiments were performed, one to measure conjunctival shrinkage, one to assess histology and inflammation 3 days after treatment, and one to measure inflammatory markers 10 days after treatment. A total of 72 female mice aged 33–41 weeks were used (26 per treatment group and 20 control) to evaluate conjunctival shrinkage, histology, inflammatory cell infiltration, tear cytokines, and expression of inflammatory genes in the conjunctiva.

Conjunctival Treatment: For experiments evaluating histology and inflammation 3- and 10-days post treatment, mice received general anesthesia with isoflurane and topical 0.5% proparacaine anesthesia. The inferior bulbar conjunctiva below the limbus in each eye was marked with a 1 mm diameter circular marker coated with gentian violet dye. The conjunctiva in the area of the mark was grasped with forceps such that the 1 mm wide marked area prolapsed above the forceps blades. This prolapsed fold of conjunctiva above the forceps blades was treated with four pulses of the near-infrared (NIR) laser in the right eye and it was touched with the thermocautery to shrink it to be flush with the surface of the forceps blades in the left eye. Mice were euthanized at post treatment days 3 and 10 to obtain tissue to evaluate the wound healing response, including the histological appearance of the treated and surrounding conjunctival tissue, density of infiltrating inflammatory cells, and expression levels of inflammatory/fibrotic mediators. Laser treatment was performed with a 1460 nm pulsed laser diode system (M1F2S22-1470.10-12C-SS5.x, DILAS, Tucson, AZ, USA) as the light source. A laser probe was attached to the handle of an angled forceps 1 cm above the platform that was used to grasp the conjunctival tissue. Laser light was delivered as a 1 × 10 mm rectangle focused on the surface of prolapsed conjunctival tissue. Four 0.2 s laser pulses (3W) delivered at 1 s intervals ensured conjunctival shrinking with minimal damage [14]. The thermocautery procedure was performed with a low-temperature battery-powered cautery (Bovie Medical, Clearwater, FL, USA). This is the device currently used to perform thermoconjunctivoplasty in the clinic.

Assessment of conjunctival shrinkage: To measure conjunctival shrinkage, thermoconjunctivoplasty was performed with the laser or cautery immediately after euthanasia by isoflurane anesthesia followed by cervical dislocation. The inferior bulbar conjunctiva below the limbus was marked with a 1 mm gentian violet dye ring. The conjunctiva was grasped with forceps and treated with either the laser or cautery. Digital images were taken of the treated areas and the width of the dye marks was measured with a digital caliper using Nikon NIS-Elements software v4.0 (Nikon, Melville, NY, USA) as a measure of treatment induced shrinkage.

Histology: Following euthanasia, eyes and ocular adnexa were excised, fixed in 10% formalin, and paraffin embedded, and 5 µm sections were cut with a microtome (Microm HM 340E, Thermofisher Wilmington, DE, USA). Sections were stained with hematoxylin/eosin or Masson-Trichrome. Sections were viewed and imaged with a microscope (Eclipse E400; Nikon) equipped with a digital camera (DXM1200; Nikon).

Confocal microscopy: Conjunctivae, including stroma and epithelium, were excised. Their immunostaining was performed with fluorescent conjugated antibodies for the neutrophil marker Ly6G-GR1 (FITC, 488 nm), myeloid marker CD11b (TRITC, 578 nm), and monocyte lineage marker Ly6C (Cy5, 647 nm). All antibodies were purchased from BD Pharmingen, San Diego, CA, USA. Confocal microscopy was performed with the Nikon A1R MP (Nikon, Melville, NY, USA) at 40× magnification. After identifying regions of interest, Z-Stack images at 40× magnification were captured using the same laser parameters for all images. Cell number and mean fluorescent intensity were measured in images using Nikon NIS-Elements software v4.0.

Tear Washings and MILLIPLEX Immunoassay: Tear-fluid washings were collected using capillary tubes, with one sample consisting of tear washing from two eyes of a treatment group (2 µL) placed into a tube containing 8 µL of BPS + 0.1% BSA and stored at −80 °C until the assay was performed. Concentrations of cytokines and chemokines (IFN-γ, IL-13, IL-1β, TNFα, IL-10, IL12(p40), IL12(p70), IP-10 (CXCL10), MIG (CXCL9), CCL2 (MCP-1)) in tear samples were assayed using a commercial Luminex MILLIPLEX Assay according to the manufacturer’s protocol (EMD Millipore Corporation, St. Charles, MO, USA). The reactions were detected with streptavidin-phycoerythrin using a Luminex 100 IS 2.3 system (Austin, TX, USA). Biologic replicate samples were averaged. Results are presented as the mean ± standard deviation (picograms per milliliter).

Conjunctival Gene Expression: Following euthanasia, conjunctiva was excised on days 3 and 10 and total RNA was extracted using a QIAGEN RNeasy Plus Micro RNA isolation kit (Qiagen) following the manufacturer’s protocol. RNA concentration was measured, and cDNA was synthesized using the Ready-To-Go™ You-Prime First-Strand kit (GE Healthcare). Quantitative real-time PCR was performed with specific murine MGB probes: IL-1b (Mm00434228), TNF-a (Mm00443260), Chemokine (C-C motif) ligand 2 (Mm00441242), Transforming Growth Factor Beta-1 (TGFβ1, Mm00441724), TGF-β2 (Mm00436952_m1), VEGF, and hypoxanthine phosphoribosyltransferase (HPRT1) (Mm00446968). The HPRT-1 gene was used as an endogenous reference for each reaction. The results of real-time PCR were analyzed by the comparative CT method, and the results were normalized by the CT value of HPRT-1.

## Figures and Tables

**Figure 1 ijms-24-05740-f001:**
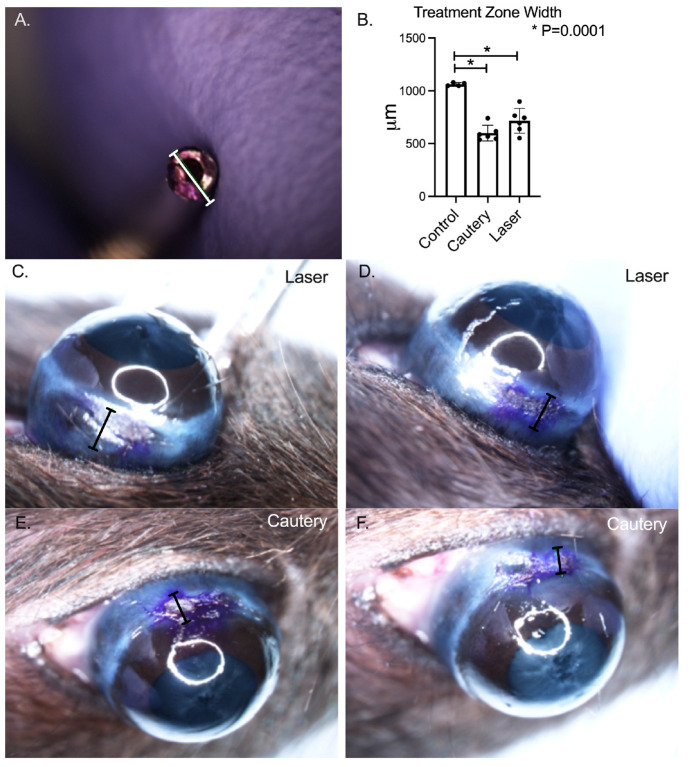
(**A**) 1 mm diameter conjunctival marker. (**B**) Mean ± SD width of marked conjunctiva (indicated with white or black lines) in untreated control and immediately following thermoconjunctivoplasty with cautery or laser (*n* = 6/group). The mean width following either treatment was significantly lower than the control (*p* = 0.0001). (**C**,**D**) Images of laser (L) treated conjunctivas. (**E**,**F**) Images of cautery treated conjunctiva. Images taken under 4× magnification.

**Figure 2 ijms-24-05740-f002:**
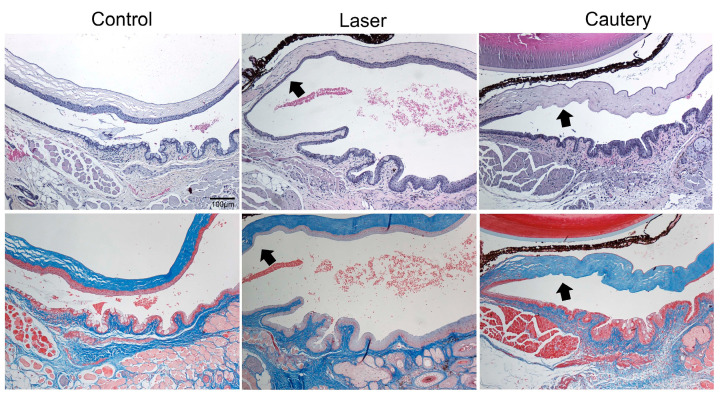
Representative tissue sections from control and day 3 laser and cautery treated eyes (*n* = 3 per group) stained with H&E (**top**) or Masson Trichrome (**bottom**) stains. Arrows designate the treatment area.

**Figure 3 ijms-24-05740-f003:**
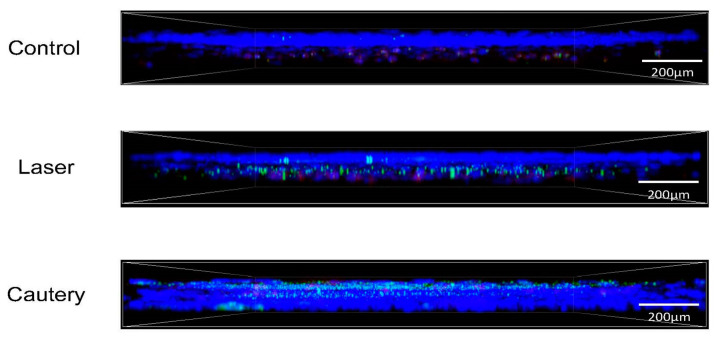
Representative Z-stack images of mouse conjunctiva in control eyes and at day 10 in the area treated with the laser or thermocautery stained with antibodies specific for Ly6G (GR-1, green) or CD11b (red). DNA is stained with Hoechst dye (blue), *n* = 3 per group.

**Figure 4 ijms-24-05740-f004:**
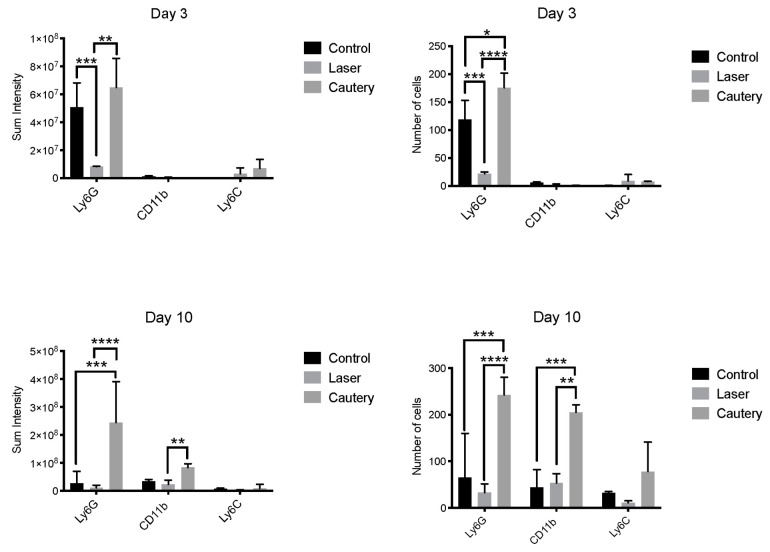
Number and sum intensity (mean ± SD) of Ly6G^+^ neutrophils, CD11b^+^, and Ly6C^+^ cells measured in immunostained whole mount conjunctiva imaged by confocal microscopy, *n* = 3 per group. *p* values: * *p* ≤ 0.05, ** *p* ≤ 0.01, *** *p* ≤ 0.001, **** *p* ≤ 0.0001.

**Figure 5 ijms-24-05740-f005:**
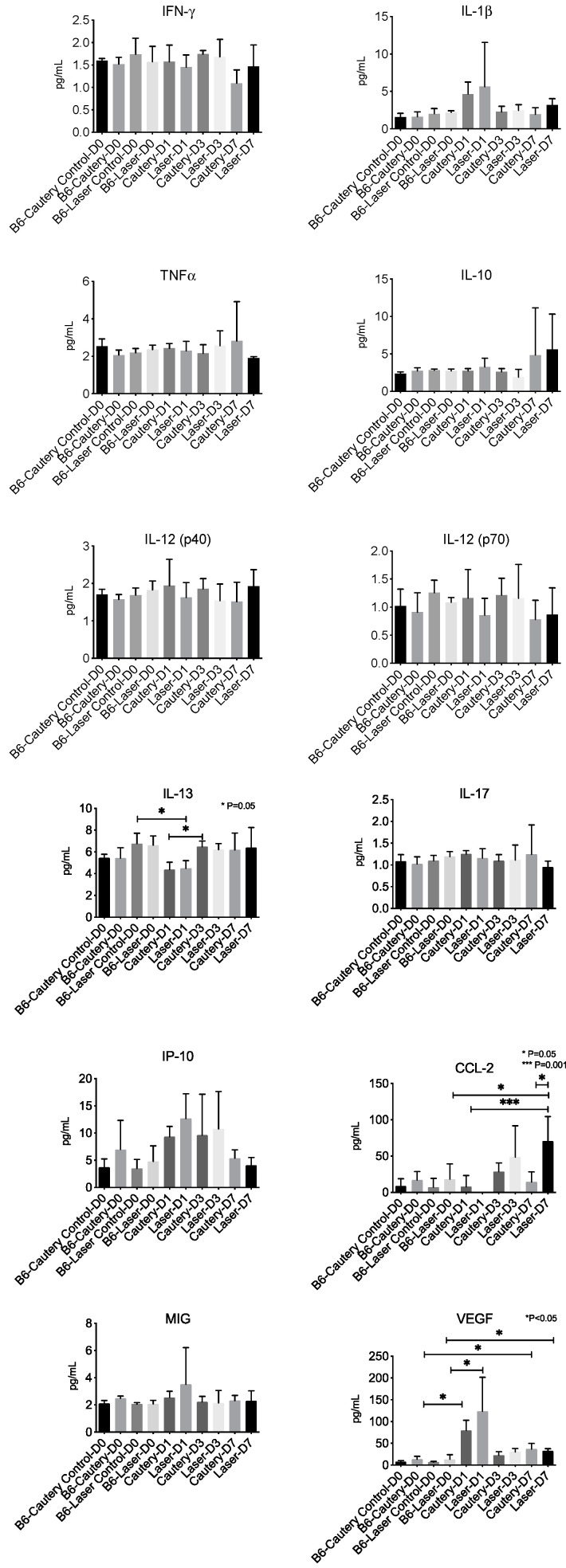
Tear concentrations of cytokines and chemokines (mean ± SD) which were measured by multiplex immunoassay in tears that were collected from untreated control eyes at D0 and eyes treated with cautery or laser at D0, D1, D3, and D7, *n* = 8 per group.

**Figure 6 ijms-24-05740-f006:**
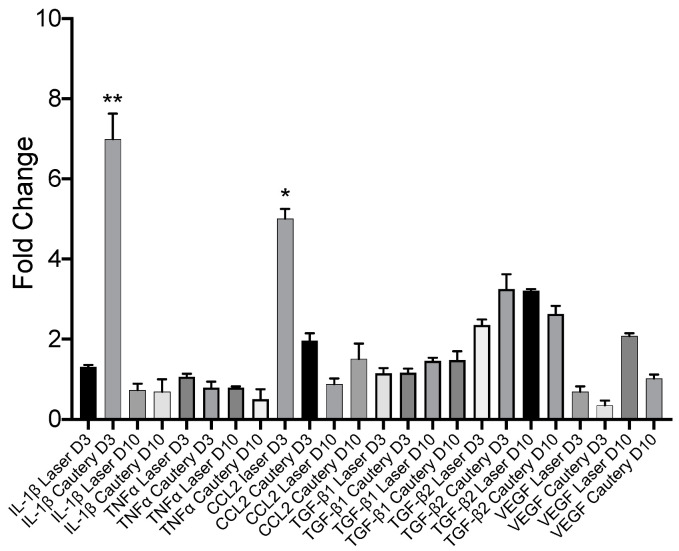
Expression of inflammatory and fibrotic genes evaluated by RT-PCR expressed as mean ± SD fold change compared to the untreated D0 control. ** *p* < 0.001 vs. IL-1b laser D3; * *p* < 0.05 vs. CCL2 cautery D3, *n* = 6 per group.

## Data Availability

All data use in the analysis is included in the manuscript.

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
