# Peer review of "Comparison of Efficacy and Inflammatory Response to Thermoconjunctivoplasty Performed with Cautery or Pulsed 1460 nm Laser"

_ijms, 2023, doi:10.3390/ijms24065740_

Round 1
Author Response
Reviewer 1
- All suggested revisions were made to the Introduction section.
- As suggested, Figure 1 was revised and the green lines were changed to white or black lines to improve visualization of the treated zone. The Fig 1 legend was revised to include this.
- Figure 4 legend was revised.
- Results Section D. Tear Cytokines. Tear cytokines were measured up to D7 and not measured at D10 because significantly lower numbers of neutrophils and CD11b+ cells between the laser and cautery groups were already observed at D3.
- Results, Section D Line 128. This was corrected to indicate there was no difference between groups at any time point.
- Results, Section D Line 133. Laser D7 compared to laser D0 was added as requested.
- Figure 5 legend was revised.
- All minor revisions were made as suggested.
- Discussion, line 168. There was no histological evidence of fibrosis 3 days following NIR laser or thermocautery, but this may not be sufficient time for a fibrotic response to be detected in tissue section. It is possible fibrosis could be detected at a later post treatment time point.
- Discussion, Line 194. The suggested revised sentence was inserted.
Reviewer 2 Report
Dear Authors,
I think that your manuscript "Comparison of Efficacy and Inflammatory Response to Thermoconjunctivoplasty Performed with Cautery or Pulsed 1460-nm Laser" can be accept in this form.
Best regards
Author Response
Reviewer 2. The reviewer noted the manuscript is acceptable for publication and did not suggest any revisions.
Reviewer 3 Report
De Souza et al. realized a very interesting communication describing the “Comparison of Efficacy and Inflammatory Response to Thermoconjunctivoplasty Performed with Cautery or Pulsed 1460 nm Laser”. I consider the manuscript very interesting but, at the same time, I suggest several revisions needed to improve the reliability and the completeness of the paper:
· The “Discussion” sections should be more updated and improved. I suggest adding data related to the involvement of oxidative stress, also focusing on vascular components, in relationship to gene expression results. The recent PMID: 32877751, PMID: 30523548, PMID: 36490268 and PMID: 36290689 could represent a substrate able to enforce the role of considered cellular mechanisms.
· The sub-chapters of “Materials and Methods” section should be better divided and evidenced.
· Are the experiments realized at least in triplicate?
· Finally, manuscript requires important English revisions and typos correction.
Author Response
- The referenced studies focus on vascular lesions in the brain and retinal degenerations. They have nothing to do with our study. I wonder if these were intended for a different manuscript.
- The subheading in the Materials and Methods section are already separate and clearly defined by assay type.
- The numbers of mice are noted in the Materials and Methods section. As noted in the figure legends, all assays had sample sizes between 3-8 in each group.
- As noted in the response to Reviewer 1, extensive revisions were made to improve sentence structure and correct typos.